# Impact of Organic and Conventional Husbandry Systems on the Gut Microbiome and Resistome in Pigs

**DOI:** 10.3390/microorganisms13092161

**Published:** 2025-09-16

**Authors:** Judith Wedemeyer, Nele Lechleiter, Andreas Vernunft, Jessica Junker, Timo Homeier-Bachmann

**Affiliations:** 1Institute of Epidemiology, Friedrich-Loeffler-Institut, Südufer 10, 17493 Greifswald, Germany; judith.wedemeyer@fli.de (J.W.);; 2Research Institute for Farm Animal Biology (FBN), Wilhelm-Stahl-Allee 2, 18196 Dummerstorf, Germany

**Keywords:** resistome, microbiome, metagenomics, animal husbandry, organic farming, pig, antimicrobial resistance genes

## Abstract

The gut microbiome of pigs is important for energy supply and impacts the animals’ health. Additionally, the microbiota can act as a reservoir for antimicrobial resistance genes (ARG). Different husbandry systems in pig farming can influence the microbiome and the overall composition of the resistome, i.e., the complete collection of ARG. In this study, pooled fecal samples were collected repeatedly in one pig farm over a period of two years. One group of animals was kept in organic husbandry conditions with access to an outdoor run, while the other group was kept according to conventional standards. Shotgun metagenomic sequencing was performed on the samples. Additionally, *E. coli* isolates were subjected to whole-genome sequencing and antimicrobial susceptibility testing. Significant differences were observed in alpha and beta diversity in the microbiome between the two husbandry systems. Families enriched in the organic group included *Prevotellaceae*, *Lachnospiraceae*, and *Cellulosilyticaceae*, while *Methanobacteriaceae* showed a higher abundance in the conventional group. In the resistome, the differences were smaller, and the dominant genes were the same in both groups. However, there was a significant difference in beta diversity. In addition, the overall frequency of ARG, normalized by 16S rRNA gene content, was on average higher in the organic group. Overall, the husbandry system shaped the microbiome and—albeit to a lesser extent—the resistome in pigs from the same farm.

## 1. Introduction

Pig meat accounted for 34% of global meat production in 2022, making pig production one of the largest branches of animal farming worldwide [1]. In conventional husbandry, the animals are kept indoors, and the stables are primarily aimed towards high productivity and biosecurity, which means high stocking densities and little space for the animals, as well as the use of slatted floors without bedding and feed aimed at maximizing weight gain [2,3]. In contrast, sustainability and improved animal welfare are of greater importance in organic farming. In the EU, Regulation (EU) 2018/848 covers the requirements for organic farming systems and demands more space for animals, additional outdoor runs, and bedding made from organic material. There are also requirements for the feed to which, for example, roughage is to be added [4]. Several positive effects on the behavior and health of pigs in such housing systems have been described [5]. In the meantime, often biosecurity remains a challenge in free-range systems [3]. An important factor that can be influenced by these different forms of husbandry is the gastrointestinal microbiome. The microbiome plays a key role in the health of pigs. In addition to its vital role in digestion, a healthy microbiome can also have positive effects on the immune system and reduce susceptibility to pathogens in the gut [6,7,8]. In addition, feed efficiency and growth performance are associated with the composition of the microbiome [9]. Various factors can impact the microbiome, such as feed, age, breed, stress, or the use of antibiotics [10]. The latter is also linked to the emergence of antimicrobial resistance in animal husbandry. Pigs were found to carry the highest loads of antimicrobial resistance genes among different livestock in Europe [11]. Resulting is the well-recognized threat of transmission of these ARG to pathogens and the spread of resistant bacteria to the environment or into the food chain [12,13]. While bacterial communities naturally harbor some ARG [14], the administration of antibiotics was shown to lead to increased resistance levels in pigs’ microbiomes [15]. However, it has been suggested that antimicrobial usage may not be the only factor that impacts the resistome, which is the total collection of ARG in the microbial population. Other factors such as stress, diet, and environmental influences could also be of importance [16,17]. In recent years, comprehensive metagenomic studies have provided a deep insight into the microbiome and resistome of pigs, showing the correlation of the composition and diversity of the microbiome with the resistome [18,19]. A few studies have so far investigated the influence of different husbandry systems on the microbiome and resistome [5,18,20,21,22]. However, the data are still limited. In particular, with regard to the resistome, there have been different observations. Gerzova et al., for example, found no significant differences between organic and conventional farms in four European countries [21], while Holman et al. found significant differences in the abundance and composition of ARG between conventionally and pasture-raised pigs [18]. Papić et al. emphasized a challenge in studies comparing farming systems [20]. They observed a significant influence of trial and farm of origin on the microbiota, which can complicate the interpretation of the influence of farming system [20]. In this study, fecal samples from pigs in organic and conventional husbandry on one farm were examined over a period of two years through shotgun metagenomics. Additional information on phenotypic resistance is provided through characterization of *E. coli* isolates. The analysis of the resistome and microbiome of pigs aims to provide information on the impact of the husbandry system and evaluate possible changes, including seasonal impact, over a period of two years.

## 2. Materials and Methods

### 2.1. Conventional and Organic Pig Barns

The samples originate from the experimental facilities for pigs of the Research Institute for Farm Animal Biology (FBN) in Dummerstorf, Germany. The experimental pig facility is divided into two types of barns. In one barn, pigs are kept according to conventional standards, and in the other barn, organic husbandry is applied. The two types of housing are in two separate barns, which are located in close proximity to each other on the premises (approximately 15 m). The organic barn had been newly built before the start of sampling and had been stocked for the first time with pregnant sows of the breed German Landrace from the conventional barn. Faecal samples were collected from German Landrace piglets that were born and reared in the respective housing system for approximately 60 days before sampling. The organic and conventional husbandry systems differed in housing conditions and feeding regimes as described below. In the organic barn, all pens are bedded with straw, and additionally, pigs have access to a straw-bedded outdoor run. In contrast, the animals in conventional housing have no access to the outdoors and are kept on flat decks made of standard plastic slatted floors. As part of the respective husbandry systems, piglets in the rearing phase also received different diets. The organic diet, composed of certified organic feedstuffs, contained more fiber (4.5% organic, 4% conventional) and was less energy-dense (organic: 13.2 MJ ME/kg; conventional: 13.6 MJ ME/kg). Details on the feed composition, including dietary ingredients, analytical components, and additives for both organic and conventional husbandry systems, are provided in the Appendix A. Data were collected on the administration of antibiotics throughout the sampling period, as well as one year prior to this study.

### 2.2. Sampling of Pig Barns

Over a period of two years, fecal samples were collected repeatedly in both barns at rearing, when the animals were around 60 days old. At the time of sampling, piglets were already weaned and kept separate from the sows. Each time, two pens were sampled per barn (in two different units of the conventional barn and in two outdoor runs of the organic barn), respectively. Fresh droppings were collected from the floor of the pens and pooled into sterile 4 mL Cryovial^®^ tubes (Carl Roth GmbH + Co, Karlsruhe, Germany). The samples were immediately frozen at −80 °C and stored until further processing. A total of 50 samples were taken from the pigs over two years. Sampling began in 12/2021 and ended in 11/2023. Details on all samples are included in the Appendix A.

### 2.3. DNA Extraction and Metagenomic Sequencing

DNA was extracted from the fecal samples using the QIAamp Fast DNA Stool Mini Kit (QIAGEN, Hilden, Germany) and following the modified protocol developed by Knudsen et al. [23]. Briefly, this protocol includes an additional step of homogenization with a TissueLyser (QIAGEN, Hilden, Germany), a higher lysis temperature of 95 °C, and an increased proportion of Proteinkinase K. Following extraction, DNA was quantified by QuBit Fluorometer (Thermo Fischer Scientific, Waltham, MA, USA) and stored at −20 °C. Library preparation and shotgun Illumina sequencing were performed by LGC Genomics GmbH (Berlin, Germany). Sequencing was carried out on an Illumina NovaSeq 6000 platform (Illumina Inc., San Diego, CA, USA), producing 250 bp paired-end reads.

### 2.4. Sequence Analysis

The AMR++ pipeline [24] was applied for identification of ARG in the metagenomes. The samples were analyzed consecutively through the pipeline, which includes steps of trimming, host removal (reference genome: GCA_000003025.6), and alignment with an ARG database, for which the MEGARES database [25] was used. In order to normalize ARG hits identified by the pipeline, 16S DNA content was quantified in each sample through the tool Metaxa2 [26]. As described in Homeier-Bachmann et al., the normalized abundance of ARG was then calculated for each sample [27]. For taxonomic classification of reads, Kraken2 [28] was used with standard settings. Sequence analysis was performed in the same way by Lechleiter et al. [29].

### 2.5. Data Evaluation

The aim was to determine the effects of the husbandry type on the microbiome and the resistome. Additionally, data were evaluated for possible changes over the time period of two years, including seasonal impact. All statistical analyses were performed in R (version 4.3.1) with R Studio (version 2023.6.2.561) [30,31]. Results were visualized through ggplot2 [32]. The package vegan version 2.6.4 [33] was used to examine diversity. Calculations were performed for both the microbiome and the ARG data set. The Shannon Index, richness, and Pielou’s Evenness were calculated as alpha diversity indices. Bray–Curtis dissimilarities between samples were calculated to evaluate beta diversity between groups. The distance values were used to perform non-metric multidimensional scaling (NMDS). Wilcoxon rank sum test was used to test for a significant difference in alpha diversity and total ARG abundance. Permutational analysis of variance (PERMANOVA) was performed to test for differences in beta diversity. *p* < 0.05 was considered statistically significant. For testing, the samples were grouped by type of husbandry and additionally by season of sampling. Meteorological spring and summer, including the months March to August, were summarized as warm season, and meteorological autumn and winter, including September to February, as cold season for this purpose to ensure a sufficient number of samples in each group. Testing for seasonal impact was performed on the whole dataset and the two groups separately. Differential abundances of taxa and ARG between the two husbandry groups were calculated using Maaslin2 version 1.15.1 [34]. Within the analysis, ARG abundances and microbial read counts were normalized by TSS and converted to a logarithmic scale. The Benjamini–Hochberg method was used to calculate adjusted *p* values, which were considered significant at <0.05. To ensure robust results, microbial taxa or ARG with a relative abundance of <0.01% or sample prevalence of <10% were excluded from the analysis of differential abundance.

### 2.6. Plating for E. coli and Antimicrobial Susceptibility Testing (AST)

Following storage at −80 °C, fecal samples of pigs were each swabbed using a sterile cotton swab, which was then streaked onto chromogenic medium CHROMagar ^TM^ Orientation (MAST Diagnostica, Oldesloe, Germany) and incubated overnight at 37 °C. To isolate possible ESBL *E. coli*, this was also performed on CHROMagar ^TM^ Orientation spiked with 2 μg/mL cefotaxime (Alfa Aesar by Thermo Fisher Scientific, Kandel, Germany). The plates were checked for supposed *E. coli* colonies based on colony morphology (red-purple colonies) and sub-cultured until pure cultures were obtained. Pure cultures were cultivated on Columbia agar with 5% sheep blood (Otto Nordwald GmbH, Hamburg, Germany) prior to performing antimicrobial susceptibility testing and DNA extraction for whole-genome sequencing. Antimicrobial susceptibility testing of the isolates was performed using the VITEK^®^ 2 COMPACT (bioMérieux, Marcy l’Etoile, France) with the software version 9.03.3 and the AST-GN96 card according to the manufacturer’s instructions. Minimal inhibitory concentration (MIC) breakpoint values were set according to the European Committee on Antimicrobial Susceptibility Testing (EUCAST) breakpoint tables for interpretation of MICs and zone diameters (EUCAST + Phenotypic DE 2023, 31.05.2023).

### 2.7. Whole-Genome Sequencing

DNA was extracted from *E. coli* isolates using the MasterPure^TM^ DNA Purification Kit for Blood, Version II (Lucigen, Middleton, WI, USA) and quantified using a QuBit Fluorometer (Thermo Fischer Scientific, Waltham, MA, USA) and stored at −20 °C. Library preparation and sequencing were performed by Seqcenter (SeqCenter, Pittsburgh, PA, USA). 2 × 150 reads were produced on a NovaSeq 6000 sequencer (Illumina Inc., San Diego, CA, USA). The sequence analysis is described in previous work [27]. In short, adapter trimming, filtering for contaminants, and quality trimming were carried out via BBDuk from BBTools v. 38.89 (http://sourceforge.net/projects/bbmap/, accessed on 19 January 2024). Through the shovill v. 1.1.0 assembly pipeline (https://github.com/tseemann/shovill, accessed on 19 January 2024) and SPAdes v. 3.15.0 [35], de novo assembly was achieved. Assembled genomes were analyzed for multi-locus sequence types (MLST) via mlst v. 2.19.0 (https://github.com/tseemann/mlst, accessed on 19 January 2024) and for antibiotic resistance genes via ABRicate v. 1.0.0 (https://github.com/tseemann/abricate, accessed on 19 January 2024). The databases used with these tools were PubMLST [36], VFDB [37], ResFinder [38], PlasmidFinder [39], BacMet [40], ARG-ANNOT [41], and Ecoli_VF (https://github.com/phac-nml/ecoli_vf, accessed on 19 January 2024).

## 3. Results

### 3.1. Antimicrobial Treatment

Antibiotics administered during the sampling period (12/2021–11/2023) were recorded. Overall, there were 59 treatments. Animals were treated individually, and no antimicrobial agents were administered via feed. The substance classes used were beta-lactam antibiotics; sulfonamide combined with trimethoprim; and, only for individual external application for skin treatment via spray, tetracycline. Beta-lactam antibiotics were used in 91.5% of the treatments. Use of antibiotics on the farm in the year prior to sampling was similar, with beta-lactam antibiotics being used in 93.3% treatments. Details can be viewed in the Appendix A.

### 3.2. Sequencing

In total, the DNA of 50 samples was sequenced at LGC. We received 12,780,045 read pairs on average (±3,362,255).

### 3.3. Microbiome

The majority of reads in both groups were assigned to the phyla Bacteroidota (organic: 42.74% ± 10.36%; conventional: 33.38% ± 6.72%) and Bacillota (organic: 34.13% ± 6.74%; conventional: 37.96% ± 4.84%). The other two dominant phyla in both groups were Pseudomonadota (organic: 11.26% ± 3.36%; conventional: 13.12% ± 2.60%) and Actinomycetota (organic: 8.43% ± 1.92%; conventional: 11.24% ± 2.16%). At the family level, *Prevotellaceae* (organic: 33.68% ± 10.04%; conventional: 23.49 ± 6.52%) and *Oscillospiraceae* (organic: 12.47% ± 5.73%; conventional: 15.37% ± 5.18%) dominated (Figure 1). *Segatella* (organic: 21.72% ± 8.52%; conventional: 11.97% ± 5.04%) and *Prevotella* (organic: 10.89% ± 2.29%; conventional: 10.40% ± 2.40%) were among the most abundant genera in both groups. In the conventional group, the percentage of reads assigned to the archaeal domain was on average higher (organic: 0.35% ± 0.072%; conventional: 0.52% ± 0.18%). In both groups, the majority of archaea belonged to the phylum Euryarchaeota (organic: 83.59% ± 8.97; conventional: 88.65% ± 4.33%). At the family level, *Haloarculaceae*, *Haloferaceae,* and *Natrialbaceae* were most abundant in both groups. *Methanobacteriaceae* occurred more frequently in the conventional group (organic: 8.73% ± 4.0%; conventional: 16.4% ± 10.94%) (Figure 2). In both groups, the microbiome remained stable over the sampling period regarding dominant taxa, and no significant difference between samples grouped by season could be identified in alpha or beta diversity.

Comparison of samples from the organic and samples from the conventional barn revealed several significant differences. The Shannon Index and Pielou’s Evenness were significantly lower in the organic group at the genus level (Shannon: W = 494, *p* = 0.00027; Pielou’s Evenness: W = 494, *p* = 0.00027 (Figure 3). PERMANOVA testing showed a significant difference in beta diversity (Bray–Curtis dissimilarity) between the groups at the genus level (R^2^ = 0.19, *p* = 0.001) (Figure 4). Maaslin2 differential abundance testing showed several taxa with different frequencies between the groups. At the phylum level, Bacteroidota were significantly enriched in the organic group. Euryarchaeota were more abundant in the conventional group. In total, 253 families showed different frequencies between the groups. These included, for example, *Prevotellaceae*, *Cellulosilyticaceae*, *Lachnospiraceae*, *Mucispirillaceae*, *Brachyspiraceae*, or *Campylobacteraceae,* which were significantly enriched in the organic group. *Methanobacteriaceae*, on the other hand, were significantly more abundant in the conventional group.

### 3.4. Resistome

On average, total ARG abundance was lower in the conventional group (conventional: 0.11 ± 0.13; organic: 0.21 ± 0.16). However, ARG abundance varied greatly between samples, ranging from 0.00081 (in sample M1200) to 0.47 in sample M1197 in the conventional group and from 0.004 (in sample M1214) to 0.65 (in sample M1211) in the organic group (Figure 5). Relative abundances of ARG were stable across the sampling period in both groups. Beta diversity was not significantly associated with season through PERMANOVA in either group. The most common ARG classes were similar in both groups. Tetracyclines dominated, but the percentage was slightly higher on average in the conventional group (70.45% ± 5.60%) than in the organic group (63.01% ± 8.16%). The *tetQ* and *tetW* genes accounted for the majority of tetracycline resistance. Three other classes were common in both groups. These were MLS (macrolides, lincosamides, and streptogramins) (conventional: 11.83% ± 3.81%; organic: 14.70% ± 3.42%), aminoglycosides (conventional 8.045% ± 2.46%; organic: 8.96% ± 2.43%), and beta-lactams (conventional: 7.08% ± 2.74%; organic: 10.08% ± 4.24). The most common gene in the MLS class was *MLS23S*, while *ant6* was a common aminoglycoside resistance gene. *Cfx* genes were the most common beta-lactam resistance genes (Figure 6).

Alpha diversity (Shannon Index) was higher in the organic group (W = 136, *p* = 0.00045) (Figure 7). There was no difference between the groups in Pielou’s Evenness or richness. There was a significant difference in beta diversity between the two groups by PERMANOVA testing (R^2^ = 0.081, *p*= 0.001) (Figure 8). Testing for differential abundance using Maaslin2 revealed two significant associations. *TetX* and *ermF* occurred more frequently in the organic group, although these genes had low mean percentage abundances of <1% in either group.

### 3.5. Cultivation

No ESBL *E. coli* were cultivated. A total of 39 *E. coli* were isolated from the 50 pig fecal samples, 21 of which were from the conventional samples, and 18 were from the samples of the organic group, respectively. Overall, the isolates of both groups showed a very similar profile of resistance genes. In both groups, all isolates carried the *mdf(A)* and *form(A)* genes. Resistance genes against tetracyclines *tet(C)*, *tet(B)*, and *tet(A)*, as well as resistance genes against aminoglycosides (*aph(6″)-ID*, *aph(3″)-Ib*, *ant(3″)-Ia*, and *aadA2*) were most frequent in both groups. Three isolates in the conventional and three isolates in the organic group carried the gene *fosA7*, which confers resistance to Fosfomycin. Some ARG were found only once in the conventional group, e.g., *blaTEM-1B*, *qacE,* or *qnrB19*. All isolates except 2 (3954, 3939) carried one or more plasmids. Twenty-four plasmid variants occurred across the isolates. The most common were IncFIB (AP001918) and Col (pHAD28). AST results showed eight isolates (20.51%) were resistant to tetracycline (see Appendix A). In addition, isolate 3995 was tested resistant to ampicillin, and isolate 3986 was resistant to trimethoprim/sulfamethoxazole. Otherwise, the isolates showed no resistance to the other substances tested, including aminoglycosides (gentamicin, neomycin).

## 4. Discussion

### 4.1. Microbiome

The microbiome and the resistome of pooled fecal samples from pigs at rearing were examined over a period of two years in two different husbandry systems. Overall, the composition of the microbiome was in agreement with previous studies of porcine gut microbiota [8,21,42]. Bacillota and Bacteroidota were the dominant phyla, followed by Pseudomonadota. Correspondingly, Holman et al. reported Firmicutes (Bacillota), Bacteroidetes (Bacteroidota), and Proteobacteria (Pseudomonadota) to be predominant overall in a meta-analysis of 20 swine gut microbiota studies based on 16S rRNA gene sequencing [42]. *Prevotellaceae*, the most frequent family in our data, was also reported as abundant in pig microbiomes across multiple studies, including those of wild boar and domestic pigs [21,43], as well as different breeds (Duroc, Landrace, Large White) [9]. Members of this family, such as the *Prevotella* species, have been associated with feed efficiency in pigs [44]. They are known to degrade plant polysaccharides, thereby providing short-chain fatty acids to their host [45]. A similar function applies to *Oscillospiraceae* [46], the second most abundant family here, which was also reported to be dominant in pig microbiota before [20]. Interestingly, the microbiota was quite stable over two years without any trend or pattern evident in dominant taxa in both groups. The first samples in the organic barn were taken from the first piglets bred in the organic barn, after it was newly built. The sows originated from the conventional barn. However, the microbiome of the organic group did not show higher similarity to samples from the conventional barn at the beginning of this study. This indicates that the influence of the housing system on the microbiome of the sampled animals at rearing was largely constant over the two years. Since animals in the organic group had access to their outdoor pens all year round, exposing them to changing climate throughout the year, we tested the data for variation in different seasons. However, the season had no significant impact on microbial diversity in either group. Seasonal effects have been observed in other domestic species, such as cattle, dairy cows, wool sheep, and horses [47,48,49,50], but they have not been extensively studied in pigs. Yet, heat stress has been shown to impact the pig microbiome [51]. Likely, the consistent feed, however, contributed to the stability of the microbiome in both groups here since diet is known to have a major influence on the microbiome [52,53]. However, it should be noted that no samples were available from the months of June or August, leaving the season with possible heat stress underrepresented. In the apparent absence of seasonal impact, interindividual variation, which is commonly observed in gut microbiomes [5,20,54], was most likely the driving factor for slight differences between individual samples in each group. Since the two barns with organic and conventional husbandry were located on the same premises and managed by the same staff, the impact of the housing system could be investigated without other possible influences of the location. Papić et al. recently showed that the farm and the trial had a significant influence on the microbiome in a study of different husbandry systems, which may complicate the evaluation of the influence of the housing system [20]. Changes occurring with increasing age from birth to the end of fattening [10,20,55,56] were also reduced as the samples were taken repeatedly at the same age of around 60 days. Comparison of the two groups showed a significant difference in beta diversity, corresponding to the results of Holman et al., who also observed differences in beta diversity in conventionally and pasture-raised pigs [18]. Surprisingly, alpha diversity indices were lower in the organic group, which was unexpected since pigs in the organic group are exposed to a more complex environmental microbiota. In contrast, alpha diversity was higher for pasture-raised pigs in Holman et al.’s study [18]. Interestingly, similar observations to ours have been made by Megahed et al., who found increased alpha diversity in pigs held on slatted floors compared to pigs in a straw-based environment [57]. *Prevotellaceae* were, on average, more frequent in the organic group here. Similarly, *Prevotellaceae* were found in higher relative abundances in piglets in enriched housing compared to conventional housing [5]. Also, several *Prevotella* species were found enriched in pasture-raised pigs [18]. Besides *Prevotellaceae*, some overall less dominant bacterial families were differentially abundant between the groups. Enriched in the organic group were *Lachnospiraceae* and *Cellulosilyticaceae*, which are notably associated with the fermentation of complex carbohydrates [58,59]. Therefore, their higher abundance likely derived from the higher amount of fiber available to the organic pigs through the feed and straw bedding. Curiously, also enriched in the organic group were some families known for their pathogenic members, such as *Brachyspiraceae* and *Campylobacteraceae* [60,61]. Of interest was also a difference in the domain archaea. Notably, there were more reads assigned to archaea in the conventional group overall. On the family level, *Methanobacteriaceae* (mostly represented by *Methanobrevibacter*) were enriched in the conventional group. *Methanobrevibacter* uses hydrogen and carbon dioxide in its metabolism, thereby producing methane [62], and has been repeatedly reported as the dominant methanogen in pigs [62,63,64]. Changes in diet, particularly an increase in fiber, were previously observed to influence the archaeal community, resulting in lower *Methanobrevibacter* abundance and reduced methane production in pigs [63,65,66]. Also, acidification of the intestinal lumen through increased production of short-chain fatty acids appears to be associated with a reduction in methane production [67]. Such mechanisms could have played a role in the organic group in this study. Together, the observations in this study indicate a significant influence on the microbiome through husbandry. On the one hand, the increased availability of plant fiber in the feed, together with the bedding material, was likely an influence. In addition, other influences could result from the form of husbandry. For example, the outdoor run with bedding allows for more movement, the expression of natural behavior such as rooting, and it has been shown that animals in enriched housing show overall more positive behavior [5]. In contrast, pigs in conventional housing show signs of chronic stress and reduced welfare [5]. Studies on rodents and humans indicate that stress can impact the microbiome [68].

### 4.2. Resistome

The resistome in pigs was previously shown to correlate with microbial composition [18,19]; therefore, we expected differences between the two groups in this study. Overall, in the resistome, relative abundances of ARG were stable across the two years in both husbandry forms. Furthermore, the results showed a high similarity in dominant ARG, but differences in total ARG abundance and diversity were observed. Both barns were located on the same farm, and animals were managed by the same staff. Likely a large influence on the resistome was the farm itself, while the differences in diversity suggest that there was a small influence of the husbandry form on the resistome. Surprisingly, the normalized abundance was higher on average in the organic group. In contrast, Holman et al. observed a significantly higher abundance in conventionally raised pigs than in pasture-raised pigs [18], and Zhou et al. also observed higher AMR levels in conventional pig farms compared to pigs in semi-free-range conditions in China [19]. In contrast, Gerzova et al. found no significant differences in abundance between conventional and organic farms in different countries [21]. Bassitta et al. quantitatively investigated the ARG content of selected resistance genes through PCR in manure from organic and conventional farms and found a higher overall quantity of ARG in the conventional farms, but for the ARGs *tetB* and *sul2*, they observed a higher abundance in the organic farms [13]. A relevant factor for the observation in this study could be that both types of husbandry were examined on the same farm. In contrast, the use of antibiotics, which is known to increase resistance [11,15,65,69], could differ between conventional and organic farms, which was discussed as a main reason for lower ARG abundances in samples from extensive Iberian swine production compared to intensive conventional systems [70]. However, in the study by Holman et al., no antibiotics were used during the study period on either farm, but they noted that antibiotics were used in the conventional facilities in the past, for example, as a short-term metaphylaxis to prevent the spread of disease [18], which could have contributed to the high difference in abundance in their study. However, in general, it should also be noted that although on average abundance was higher in the organic group, there were strong differences between individual samples in both groups. Therefore, further studies are needed to verify and better understand this observation. In both groups, Tetracyclines were the dominant class of ARG, mostly represented by the genes *tetW*, *tetQ*, and *tetO*. These genes have been found dominant in swine resistomes repeatedly [19,71,72,73] and have also been linked to bacterial species considered commensal members through analysis of metagenome-assembled genomes (MAG) [74]. *TetW* and *tetQ* encode for ribosomal protection proteins [75]. The genes have been found in Gram-positive and Gram-negative bacteria and are associated with mobile genetic elements in bacterial chromosomes, which may facilitate their wide distribution in gut microbial communities [76,77,78]. Besides Tetracyclines, MLS and aminoglycosides are found to be abundant classes in several metagenomic studies on pig resistomes, as reviewed by Ma et al. [16,74], which aligns well with our data. Beta-lactam resistance genes were mainly represented by the *cfx* gene group, which encodes for a class A beta-lactamases [25]. *CfxA2* was found in several MAG by Holman et al. [74]. Additionally, *cfxA2* was correlated with different *Prevotella* species in the study of Guo et al. [73]. *Prevotellaceae* were a dominant family in both groups, which could be an indication of the relative abundance of these genes in our data. Alpha diversity was higher in the organic group, and the resistomes were significantly different regarding beta diversity, which suggests that although there was a large proportion of shared ARG between the two groups, the type of husbandry seemed to have influenced ARG composition. This compares to the observations on ARG diversity for pasture-raised and conventionally raised pigs [18]. Testing for differentially abundant ARG yielded only a few significant results. Only the genes *tetX* and *ermF* were significantly enriched in the organic group. Notably, these genes had overall low abundances. Interestingly, *tetX* and *ermF* were recently found on the same contig in a swine gut-derived *T. succinifaciens* MAG [74]. In the study of Holman et al., there were 155 differentially abundant ARG between conventional and pasture-raised pigs, which is more than in this study [18]. Possibly, the exposure to environmental microbes could be greater on a pasture compared to a straw-bedded paddock, which could result in greater influence on the resistome through environmental bacteria. Antimicrobial usage was considered low here since on the farm, only individual animals were treated occasionally during the study period. In contrast, metaphylactic administration of antibiotics through feed is sometimes practiced in pig farming and would result in an overall higher amount of antibiotics used [79]. Here, beta-lactams were used in 90% of cases, while tetracyclines were rarely used, and only in the form of external sprays for skin treatment. Remarkably, however, the proportion of beta-lactam resistance in the microbiome was low, especially in comparison to tetracyclines. ARG of the classes MLS and aminoglycosides were also more frequent as described above. Mencía-Ares et al. also observed no significant correlation in the resistome in farms that mainly used beta-lactam antibiotics [22,70]. Similarly, van Gompel et al. observed positive associations for respective ARG abundance and the use of macrolides and tetracyclines but not for beta-lactams [15]. Overall, the most common resistances here do not seem to be related to recent antimicrobial treatment.

In addition to the metagenome, *E. coli* was isolated from the fecal samples of pigs, and resistance phenotypes and genotypes were analyzed. ARG conferring resistance to Fosfomycin were present in the pig metagenome, albeit at relatively low frequency. Whole-genome sequencing of *E. coli* isolates revealed that one gene in this class (*fosA7*) was carried by several *E. coli*. All isolates carried the *mdfA* gene, which was previously found in high frequencies in *E. coli* isolates from dairy farms [80] and pigs [81]. Overall, the proportion of resistance in the isolates was low compared to surveillance data of commensal *E. coli* in pigs [82]. In particular, beta-lactam resistance was detected at a higher prevalence in different farms from the same region, as in the farm investigated here [83]. One reason for this could be the overall low use of antibiotics on the farm, as discussed above. However, it is interesting to note that the most frequent phenotypic resistance of the isolates to tetracyclines coincides with the largest proportion of ARG in the metagenome, which was in the tetracycline class.

### 4.3. Limitations

For metagenomic studies, it has previously been shown that sequencing depth has a major impact on the ability to recognize the full range of ARG and microbial taxa [84,85]. Zaheer et al. showed a significant increase in ARG diversity from 26 to 117 million reads [85]. With an average of 12.8 million read pairs, the sequencing depth in this study was low. Especially in the resistome, rare ARG might be underrepresented.

## 5. Conclusions

In this study, the microbiome and resistome of pigs in two housing systems on one farm were analyzed over two years. Microbial and ARG composition were stable over the period studied. Several bacterial and archaeal taxa were significantly associated with the different husbandry systems. In the microbiome, increased fiber content from the feed and the available bedding material appeared to have a particular influence. In the resistome, the differences appeared to be smaller than in studies examining housing systems on different farms, which could indicate a large influence of the farm itself. Nevertheless, the differences in ARG composition between the two groups emphasize a multifactorial event in the reservoir formation of ARG in pigs, in which management factors play a role in addition to the use of antibiotics.

## Figures and Tables

**Figure 1 microorganisms-13-02161-f001:**
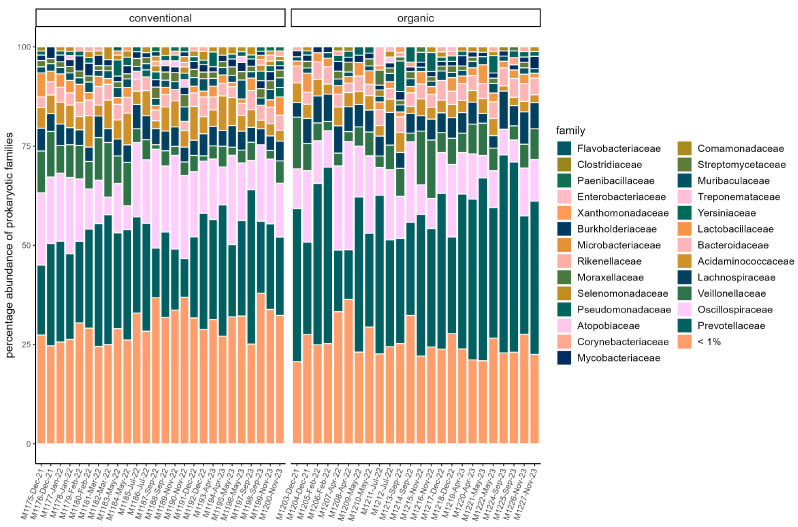
Relative abundance of reads assigned to microbial families in each sample. Families with a percentage of <1% are pooled. Samples are faceted by husbandry (conventional, organic). Samples are ordered chronologically from the first point of sampling to the last for each husbandry form, respectively. On the *x*-axis, labels include sample ID, month, and year of sampling.

**Figure 2 microorganisms-13-02161-f002:**
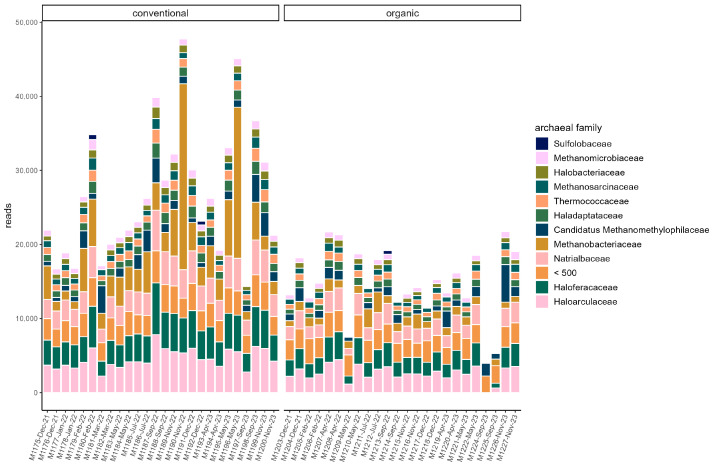
Read counts of archaeal families in each sample. Families with a read count of <500 are pooled. Samples are ordered chronologically from the first point of sampling to the last for each husbandry form, respectively. On the *x*-axis, labels include sample ID, month, and year of sampling.

**Figure 3 microorganisms-13-02161-f003:**
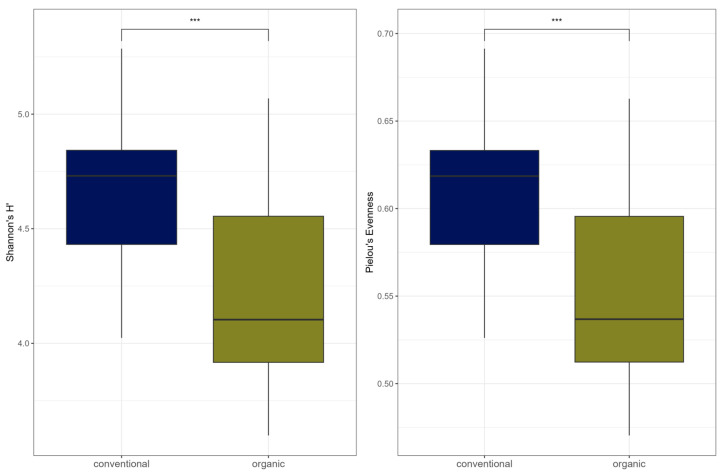
Alpha diversity (Shannon Index and Pielou’s Evenness) for microbial genera. In the boxplots, the central horizontal line is the median, the lower and upper hinges correspond to the 25th and 75th percentiles, and the whisker is 1.5 × IQR (interquartile range). Significant differences are indicated by brackets and asterisks (*** *p* < 0.001).

**Figure 4 microorganisms-13-02161-f004:**
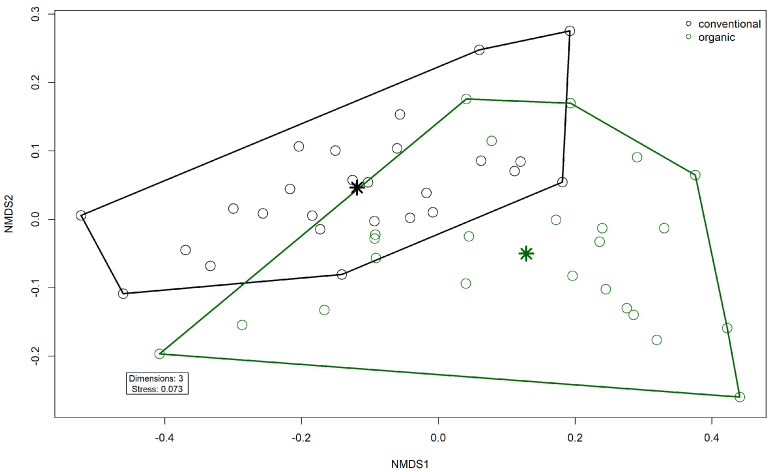
Non-metric multidimensional scaling (NMDS) plot of the Bray–Curtis dissimilarities of microbial genera in the fecal microbiomes. Colors indicate husbandry form (conventional, organic). Asterisks indicate group centroids in NMDS space.

**Figure 5 microorganisms-13-02161-f005:**
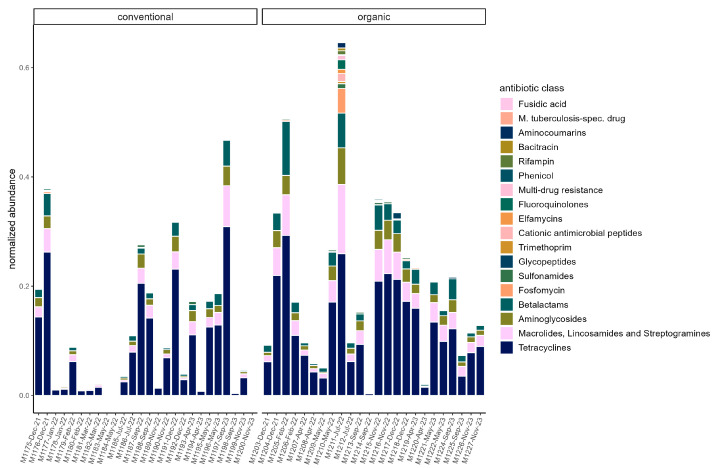
Normalized abundance of ARG summarized by antibiotic class in each sample. Samples are faceted by husbandry (conventional, organic). Samples are ordered chronologically from the first point of sampling to the last for each husbandry form, respectively. On the *x*-axis, labels include sample ID, month, and year of sampling.

**Figure 6 microorganisms-13-02161-f006:**
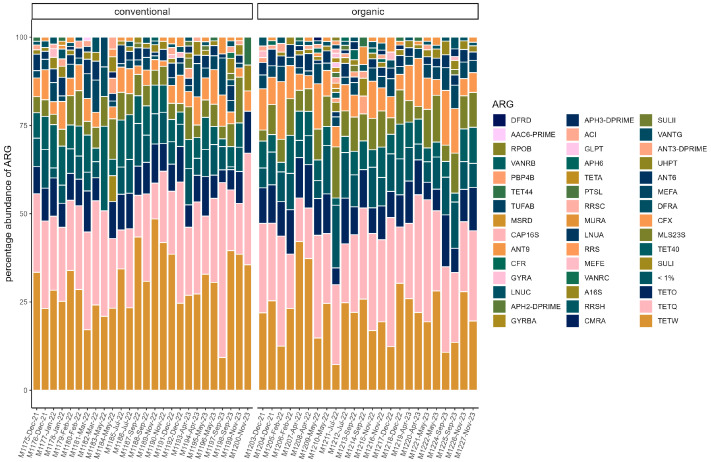
Relative abundance of ARG in each sample. ARG with a percentage of <1% are pooled. (conventional, organic). Samples are ordered chronologically from the first point of sampling to the last for each husbandry form, respectively. On the *x*-axis, labels include sample ID, month, and year of sampling.

**Figure 7 microorganisms-13-02161-f007:**
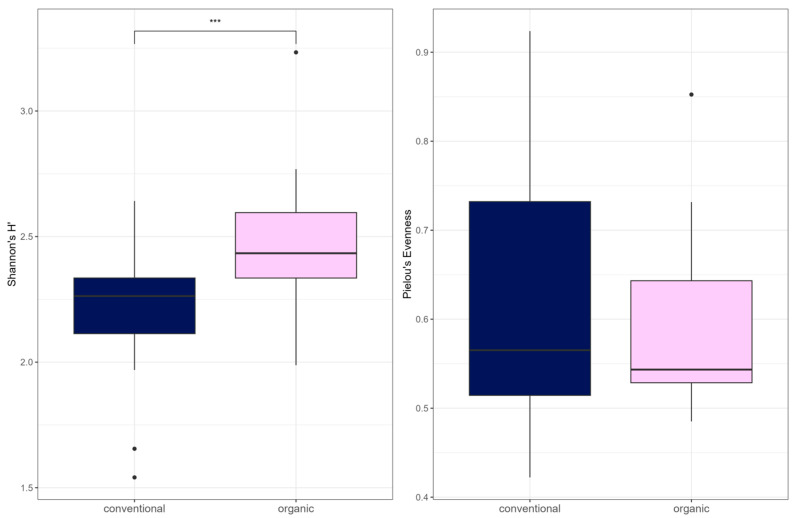
Alpha diversity (Shannon Index and Pielou’s Evenness) for ARG. In the boxplots, the central horizontal line is the median; the lower and upper hinges correspond to the 25th and 75th percentiles, and the whisker is 1.5 × IQR (interquartile range). Data points beyond the whiskers (shown as dots) represent outliers. Significant differences are indicated by brackets and asterisks (*** *p* < 0.001).

**Figure 8 microorganisms-13-02161-f008:**
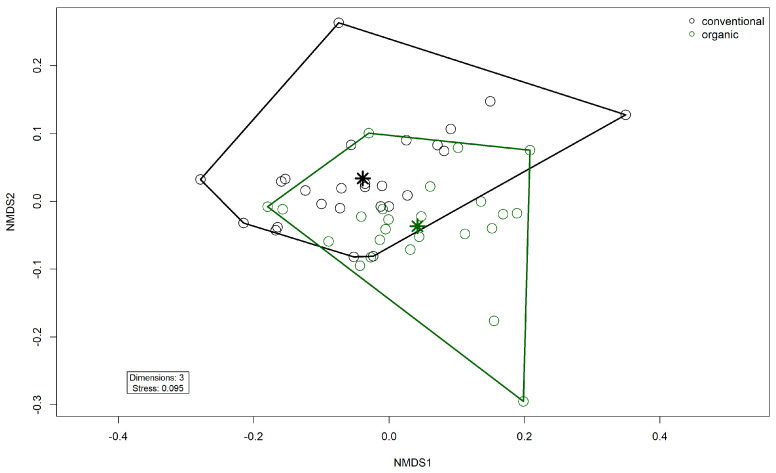
Non-metric multidimensional scaling (NMDS) plot of the Bray–Curtis dissimilarities of ARG (gene level) in the fecal microbiomes. Colors indicate husbandry form (conventional, organic). Asterisks indicate group centroids in NMDS space.

## Data Availability

The datasets generated and analyzed for this study can be found in the European Nucleotide Archive repository under the project accession number PRJEB94130.

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
