# Peer review of "Impact of Organic and Conventional Husbandry Systems on the Gut Microbiome and Resistome in Pigs"

_microorganisms, 2025, doi:10.3390/microorganisms13092161_

Round 1

Reviewer 1 Report

Comments and Suggestions for Authors

This study investigated the Impact of Organic and Conventional Husbandry Systems on the Gut Microbiome and Resistome in Pigs, the manuscript is organized in line with the guidelines and the results are presented in correct ways. However, the authors should carefully address the following concerns.

Introduction:
Line 14: it seems that some words are missing between “within” and “(resistome)” .
Line 67-68: reference citation is missing.

Materials and methods:
The parity and breeds of sows should be provided.
Organic husbandry system should be defined in details. 
Use italics for the names of microbiota except for phylum level.
Check grammar issues, such as “is” and “are” in line 81, etc.
Lines 89-90: the feed is offered to sows or suckling pigs?, please specify it. In addition, it is well known that different diets (especially diet ingredients) have significat impact on microbial composition, why the authors use different types of diet to feed animals?. Are the suckling pigs offered the same creep feed during rearing period?, if indeed, information of the creep feed and feeding method should be provided.
Lines 90-93: the comma in “13,2 MJ/kg” and “13,6 MJ ME/kg” is incorrect. What kinds of data have been collected?
Lines 96-104: more detailed information are needed for the sampling of dropping especially in organic barn, such as sampling was performed inside the organic barn, inside the organic barn and outside run?, are all sows in organic barn housed in crates or loose?, how to prevent the fresh droppings of piglets from mixing with sow’s feces?

Discussion
Line 331: the sentence “wool sheep and horses. [47–50] but have not been extensively studied in pigs.” should be corrected.
Line 334: the comma between “noted” and “that” should be deleted.
Line 385: a comma is needed between “[18, 19]” and “therefore”.

References
Standardize the format of references, such as the journal format of reference No.7

Reviewer 2 Report

Comments and Suggestions for Authors

Abstract

L26-27: The sentence 'The study analyzes the impact of two different husbandry systems on the microbiome and resistome of pigs from the same farm' is more or less reemphasising the aim of the study, which has already been implied at the beginning of the abstract. Rather, replace the sentence with a takeaway message as a conclusion.

Introduction

L32: 'Pig production represents a large proportion of animal farming worldwide'. This sounds vague. Specify the proportion of pig production to animal farming worldwide in numbers for clarity.

Materials and Methods

L83-84: Quantify the proximity in numerical value to improve reproducibility of the study. For clarity, include information on how long the animals have been acclimatized or stayed in the two systems before faecal sample collection started. 

Discussion

L309-311: Include references to support the sentence.

L312-315: Include the species and breed the gene sequencing was carried out on.

Reviewer 3 Report

Comments and Suggestions for Authors

The work presented is undoubtedly interesting in the field of pig farming, even if the authors themselves indicate the limitations of the metagenomic study they carried out. Nonetheless, interesting insights are provided regarding the impact of different housing systems on the distribution of antibiotic resistance microbiota and genes.

Some questions:

1) Why did you also introduce the diet variable into your study? Wouldn't it have been more linear to introduce only the organic vs conventional environment initially?

2) Why did you choose to evaluate antibiotic resistance and perform WGS only on E. coli strains? These are responsible for typical diarrheal forms in piglets, but there are other related pathogens (e.g., Salmonella)

3) In lines 152-153, you wrote "Fecal samples of pigs were each swabbed using a sterile cotton swab which was then 152 streaked onto chromogenic medium ...", did you run the isolation before or after freezing the sample for subsequent sequencing?

4) In lines 159-160, what kind of other characterisation was performed?

5) In your opinion, why have you found a high presence of tetracycline resistance genes?

In line 190, cancel "the"

Please, in the text, put in italic all the bacterial Genus (line 204) 

Round 2

Reviewer 1 Report

Comments and Suggestions for Authors

No further comments